# The Initial Stage of Climatic Aging of Basalt-Reinforced and Glass-Reinforced Plastics in Extremely Cold Climates: Regularities

**DOI:** 10.3390/polym16070866

**Published:** 2024-03-22

**Authors:** Anatoly K. Kychkin, Anna A. Gavrilieva, Aisen A. Kychkin, Irina G. Lukachevskaya, Mikhail P. Lebedev

**Affiliations:** 1Siberian Branch of the Russian Academy of Sciences V.P. Larionov Institute of Physical and Technical Problems of the North, 1 Oktyabrskaya Str., 677000 Yakutsk, Russia; gav-ann@yandex.ru (A.A.G.); icen.kychkin@mail.ru (A.A.K.); mirkin1611@gmail.com (I.G.L.); 2Siberian Branch of the Russian Academy of Sciences Federal Research Center <<Yakut Scientific Center SB RAS>>, 2 Petrovskogo Str., 677000 Yakutsk, Russia; m.p.lebedev@mail.ru

**Keywords:** climatic resistance, deformation–strength characteristics, basalt-reinforced plastic, glass-reinforced plastic

## Abstract

Detailed analyses of the reasons for changes in the mechanical parameters of fiberglass exposed to different climatic zones have been made available in the literature; however, such detailed studies of basalt plastic do not yet exist. It is possible to make reasonable conclusions on the climatic resistance of reinforced plastics by monitoring the deformation–strength characteristics in combination with fractographic and DMA analyses of the solar- and shadow-exposed parts of the plastics; additionally, one can conduct analyses of the IR spectrum and the moisture sorbtion kinetics. As a starting point for the climatic aging of polymer composite materials, it is necessary to accept the time of exposure in which the maximum values of the elastic strength properties of polymeric materials are achieved. Based on the results of the DMA analysis, it was found that, unlike basalt-reinforced plastics (where the material is post-cured exclusively at the initial stage of the exposure), in glass-reinforced plastic, a process of destruction occurs. The formation of internal stresses in the material and their growth were determined through observing the duration of climatic exposure. The formation of closed porosity, depending on the duration of exposure, can be assessed using the values of the increase in the average moisture content. A set of experimental studies has established that glass-reinforced plastics are subject to greater destruction under the influence of a very cold climate than the basalt-reinforced plastic.

## 1. Introduction

Currently, with development in Arctic regions, the attention of researchers has increasingly been drawn to the problem of the aging of polymer composite materials (PCMs) in cold climates. The relevance of the problem is steadily increasing with the increase in the use of PCMs in aircraft manufacturing, space technology, shipbuilding, the automotive industry, construction, and various branches of mechanical engineering. The basic properties of PCMs directly depend on the composition of the components, their quantitative ratios, and the strength of the bond between them. At the same time, the creation of new materials with specified properties for certain operating conditions requires clarification of the patterns of destruction and the determination of basic principles; the aim is to slow down the dynamics of the destruction process based on a comparative assessment of the complex of properties held by the materials in their original state and after exposure to aging factors. The use of these materials is complicated by the fact that, in both domestic and global practice, there is no strict scientific basis for the reliable prediction of the durability of PCMs for long periods of operation. Currently, detailed analyses of the changes in the mechanical parameters of glass-reinforced plastics exposed to different climatic zones are available in the literature [1,2,3]; however, such detailed studies for basalt-reinforced plastic do not yet exist.

The problem of ensuring high-performance properties for PCMs is especially clearly revealed when products are used in areas with extreme natural and climatic conditions. This determines the undoubted relevance of carrying out full-scale climatic tests of reinforced plastics and monitoring their deformation–strength characteristics.

PCM’s thermal and humidity effects, accompanied by simultaneous or alternate UV irradiation, leads to a significant increase in the likelihood of microcracks forming in PCMs [4,5,6,7,8,9,10]. As a result, the reinforcing filler is exposed and destroyed, reducing the ability of the materials to bear the load. Therefore, monitoring the microstructure of the solar- and shadow-exposed sides of reinforced plastics is necessary.

At the initial stage of climatic aging, it is possible to increase the stability indicators of PCMs, including those which come as a result of the post-curing of the polymer matrix [11,12]. Therefore, it is necessary to monitor indicators characterizing the cross-linking of the polymer matrix—the glass-transition temperature.

When PCMs are exposed in cold climates, the processes of plasticization, swelling, hydrolysis, post-curing, and destruction of their polymer matrices under the influence of temperature, humidity, and solar radiation are less pronounced than they are in the tropics and subtropics. During long-term low-temperature seasonal and daily cycling, internal stresses arise in the exposed samples due to differences in the linear thermal expansion coefficients for the reinforcing fibers and polymer matrices. These internal stresses contribute to the growth of microcracks, their merging, and the formation of macrodamages. Capillary moisture condenses into these macrodamages, which can turn into a solid phase at temperatures below 0 °C and be a source of additional internal stresses that cause the formation of new damages and cause a decrease in the strength of PCMs [13]. Therefore, it is necessary to monitor indicators characterizing the internal stress gradient, i.e., the elastic modulus and the loss modulus of the solar- and shadow-exposed sides of reinforced plastics.

A necessary condition for studying the climatic aging of polymer materials is not only the determination of moisture content but also the analysis of the kinetics of moisture transfer during the exposure of samples in open climatic conditions. According to [14,15], the moisture diffusion coefficient and the limited moisture content are sensitive indicators of physicochemical transformations in the surface layers of polymer materials when they are exposed to climatic conditions.

Chemical resistance of epoxy matrices and reinforcing fillers to the effects of extremely cold climates can be determined by changing the IR spectra of reinforced plastics.

Thus, the purpose of the present paper is to identify the regularities of the initial stage of climatic aging in the extremely cold climate of Yakutsk; this study used two materials—basalt-reinforced plastic (BRP) and glass-reinforced plastic (GRP)—and monitored their deformation–strength characteristics and analyzed the changes that occurred in the surface exposure of the reinforcing fabric, the glass-transition temperature, the elastic modulus, and the loss modulus. Specifically, these aspects were measured during the thermal heating of the solar- and shadow-exposed parts of reinforced plastics; the kinetics of moisture transfer were assessed in the absorption region for infrared radiation in reinforced plastics (RPs).

## 2. Materials and Methods

The initial binder comprises 56% wt. epoxy-diane resin ED-22 (Kukdo chemical Co., Ltd., Seoul, Korea), 42.5% wt. iso-methyltetrahydrophthalic anhydride iso-MTPHA (JSC Sterlitamak Petrochemical Plant, Sterlitamak, Russia), 0.8 % wt. an accelerator 2,4,6-tri-N/N/dimethylaminomethyl phenol Agidol 53 (JSC Sterlitamak Petrochemical Plant, Sterlitamak, Russia).

The ester group—formed as a result of curing (Figure 1)—is resistant to the action of organic and many inorganic acids, but is destroyed by alkalis; however, the thermal stability and electrical insulating properties are higher than they are when amine hardeners are used.

Basalt fabrics BT-11/1 (LLC Factory of technical fabrics, Vladimir, Russia) with high chemical (in acidic and slightly alkaline or cement media) and thermal resistance are made from basalt roving or twisted threads using a sizing agent compatible with epoxy resin.

Fiberglass fabric Ortex 560 (LLC BauTex, Gus-Khrustalny, Russia) is produced using German technologies. Silane lubricants are used in the production of fabric, which ensure the compatibility of Ortex fabrics with polyester, vinyl ester, epoxy and phenolic resins.

The objects of study were samples based on epoxy binders with a thickness of 5 mm, BRP made from 15 layers of BT-11/1 basalt fabric with twill weaving, and GRP made from 13 layers of Ortex 560 fiberglass fabric with plain weaving.

To study the influence of climatic factors on the properties of laminated plastics, samples were exposed at the test site in Yakutsk at the shared core facilities of the Federal Research Center, Yakutsk Science Center SB RAS in accordance with GOST 9.708-83 [16].

The tensile and bending strength limits of the samples were determined by testing polymer composites in tension regulated by GOST 32656-2014 [17], in three-point bending regulated by GOST 25.604-82 [18] and GOST 4648-71 [19].

The surface microstructure of the samples was determined using a JSM-7800F scanning electron microscope (JEOL, Tokyo, Japan) at an accelerating voltage of 2 kV in a vacuum environment.

The glass-transition temperature was determined by dynamic mechanical analysis using a DMA 242C instrument (NETZSCH, Selb, Bavaria). To test the samples, a three-point bending holder with a span length of 40 mm between supports was used. Tests were carried out in accordance with GOST R 56753-2015 [20] at a temperature range from 25 to 150 °C with a heating rate of 5 °C/min, amplitude 10 μm, and frequency 1 Hz, in an argon environment (gas flow rate 50 mL/min).

The kinetics of moisture saturation in the samples was determined in accordance with GOST R 56762-2015 [21]. Relative humidity and air temperature were kept at 68% and 23 °C. Moisture desorption from the samples was determined in accordance with GOST 29244-91 [22]. The samples had a square base 50 × 50 mm with unprotected boundaries.

The composition and chemical structure of the samples were determined using a 7000 FT-IR Fourier transform IR spectrometer (Varian, Santa Clara, CA, USA). Spectra were taken at room temperature for samples with dimensions of 15 × 5 × 5 mm in the frequency range 400–4000 cm^−1^. The results were processed using the open-source Essential FTIR software 3.50 build 225 (Operant LLC, Monona, WI, USA).

## 3. Results and Discussions

### 3.1. Monitoring Elastic Strength Properties

To assess changes in the mechanical properties of the reinforced plastics (RPs) exposed to extremely cold climates in this study, a relative retention coefficient, $k_{R}$, is introduced according to the following formula: (1)$$k_{R}=R_{t}/R_{0},$$
where $R_{t}$ is the tensile or bending strength of the sample, *t* is the exposure period, and $R_{0}$ is the tensile or bending strength of the sample in the unexposed state.

Table 1 presents the persistence indicators $k_{R}$ (Equation 1) of RPs after 2 and 4 years of exposure at open atmospheric stands in Yakutsk.

According to Table 1, there is an increase in the physical and mechanical properties of RPs after 2 years of exposure. After 4 years of exposure, the values of the tensile strength retention coefficients of RPs decrease by 11%, the values of the bending strength retention coefficients of BRP increase by 31%, and the values for GRP decrease by 9%. At the same time, relative to 2 years of exposure, after 4 years of exposure, the tensile strength characteristics of BRP are reduced by 15% and for GRP they are reduced by 22%; the bending strength of BRP is reduced by 12% and for GRP this is reduced by 47%. Thus, to assess the climatic resistance of PCM, it was proposed that the time at which the maximum values of the elastic strength properties of polymer materials are achieved should be taken as a starting point ($R_{0}=R_{2years}$).

### 3.2. Monitoring the Surface Microstructure of RPs

The ultraviolet component of solar radiation is considered to be the most important factor in the climatic impact on PCMs because it initiates photochemical reactions in the polymer matrix; these lead to the irreversible destruction of the material. Because the polymer matrix binds and transfers load to the reinforcing fibers, damage to this critical component can adversely affect the overall mechanical properties of PCMs and limit their service life.

UV radiation initiates chain scission reactions, causing the end fragments of the chain to occupy more volume than the non-end fragments, causing stresses that lead to the formation of cracks. Moreover, if the decomposition products are volatile or gaseous, pores and depressions may form. The above areas initiate further destruction under the influence of moisture sorption and temperature cycling.

The microstructures of the front surfaces of the BRP and GRP samples from the side of solar exposure are shown in Figure 2.

The general structure of unexposed RP samples (Figure 2a,d) is characterized by the presence of technological pores. When the structure of the control sample of RP is enlarged, it is clear that the binder polymer covers the surface evenly. The general structure of the samples after exposure for 2 years (Figure 2b,e) is characterized by the formation of cracks in the polymer matrix; the thickness of cracks for BRP varies from 1 to 8 μm and the thickness of cracks for hydraulic fracturing varies from 1 to 3 μm. When magnified, it is clear that active crack growth forms around pores. There are also areas of bare basalt fibers, ranging in size from 200 to 500 microns. The general structure of RP samples after exposure for 4 years (Figure 2c,f) is characterized by an increase in exposed areas of reinforcing fabric; for BRP, the dimensions of exposed fibers range from 600 to 1000 μm; for GRP, the dimensions of exposed fibers range from 1000 μm.

Figure 3 shows the microstructure of the reverse side of the RP samples from the side that was not exposed to sunlight.

On the structure of the unexposed RP samples (Figure 3a,d), technological pores are observed, ranging in size from 30 μm to 1.5 mm; for the BRP sample, the binding polymer envelops all the fibers, in some places with a thin layer; for the GRP sample, the binding polymer does not envelop all fibers; in some places, there are empty spaces. The general structure of RP samples after exposure for 2 years (Figure 3b,e) is characterized by the formation of cracks in the polymer matrix, which also spread to the fibers, leading to their exposure. On the general structure of RP samples after exposure for 4 years (Figure 3c,f), the formation of cracks that connect the pores and lead to the destruction of the polymer is observed, leading to the exposure of the reinforcing fabric.

Figure 4 shows the microstructure of the longitudinal sections of BRP and GRP samples after exposure.

As a result of fractographic studies, we found evidence for the occurrence of destructive processes in the surface layers of composites facing the sunny side and the detachment of the polymer matrix from the reinforcing material, depending on the exposure time. Moreover, in GRP, this effect is observed already from the second year of exposure (Figure 4e,f); in BRP, detachment is observed from the 4th year (Figure 4c). From the age of 4 years, the delamination of the joint increases and additional cracks appear in the polymer matrix (Figure 4c,f).

Under the influence of climatic factors, the destruction of the polymer matrix is observed on the surface of RP samples; the greatest change in the structure occurs on the side facing the sun. With further exposure, areas of the reinforced fabric of the RP samples will be exposed to climatic factors.

### 3.3. Monitoring of Thermomechanical Parameters of Solar- and Shadow-Exposed Parts of RPs

The glass-transition temperature of the polymer matrix ($T_{g}$) is one of the sensitive indicators of the climatic aging of PCMs. DMA methods are widely used to determine this indicator [23,24,25].

RP samples were divided into solar- and shadow-exposed parts. Figure 5 and Figure 6 show the temperature dependencies of the elastic modulus, the loss modulus, and the mechanical loss tangent of the samples of the solar- and shadow-exposed sides of the BRP and GRP at the stages of exposure: unexposed, exposed for 2 years, and exposed for 4 years.

The original samples were reheated above the glass-transition temperature ($T_{g}+40$ °C). Repeated heating led to the partial destruction of the material and a decrease in the elastic modulus and glass-transition temperature. Thus, DMA curves show the actual glass-transition temperature and moduli.

The results of the dynamic mechanical analysis show that, at the initial stage of exposure in the BRP layers, the curing of the polymer matrix occurs, which is accompanied by a shift in the peak of the loss modulus towards higher temperatures. In the 4th year of exposure, the matrix in the polymer material is structured this is confirmed by a narrowing of the temperature range of the devitrification of the material. In this case, stresses relax on the illuminated surface, and accumulate on the shadowed surface; this is accompanied by a decrease in the intensity of the transition at the dynamic loss modulus. At the initial stage of exposure, only post-curing has occurred in the material of the BRP samples; meanwhile, in the GRP samples, both the post-curing process and the destruction process have occurred. These are accompanied by a decrease in the storage modulus by more than 50% and a decrease in the loss modulus of more than 2.5 times. In addition, a bifurcation of the peak of this loss modulus is observed; this means that the material in the GRP sample has undergone destruction. The change in the dynamic modulus of the elasticity of BRP after exposure for 4 years is no more than 30%; this indicates the climatic resistance of BRP; GRP is more susceptible to changes in its physical and mechanical properties under the influence of solar radiation.

Table 2 and Table 3 show the values of the characteristic temperatures of the DMA analysis of the solar- and shadow-exposed sides of the BRP and GRP.

According to GOST R 56753-2015 [20] the inflection point on the elastic modulus curve is taken as the glass-transition temperature. Figure 7 graphically shows the change in glass-transition temperature after the climatic testing stages.

The obtained result confirms the formation of a gradient of internal stresses in the material and its growth depending on the duration of the climatic impact. An increase in the internal stress gradient occurred with a significant difference in the glass-transition temperatures of the solar- and shadow-exposed sides of the RP samples. The internal stress gradient in the GRP exposed for 2 years is higher than the internal stress gradient in the BRP exposed for 2 years.

### 3.4. Monitoring the Kinetics of Moisture Absorption and Desorption

The moisture content in the material was determined as the average increase in the mass of three samples to the mass of the corresponding dry sample in percentage terms according to the following formula: (2)$$M=\left(W_{i}-W_{0}\right)/W_{0},$$
where $W_{t}$ is the mass of the sample, g; $W_{0}$ is the mass of the dry sample, g; *i* is the mass measurement number.

Figure 8 shows the experimental values of moisture sorption and desorption in BRP and GRP both unexposed and after exposure, depending on the square root of time. The sorption kinetics are characterized by two stages: the first is continuous linear growth with a relatively small sampling error; the second is a jump followed by a decrease, stabilization, and again an almost linear increase in moisture content with a relatively large sampling error. The desorption stage is characterized by the fact that, at the beginning, there is a linear decline. Thus, the first stage of sorption and desorption of moisture was approximated by the Fick diffusion model (*D* is the Fick diffusion coefficient and $M_{1}$ is the limiting moisture content of the first stage). The second stage was approximated by a jump in moisture content according to
(3)$$\Delta M=\sum_{i}\left(M_{i}-M^{F}\left(t_{i}\right)\right)/N,$$
where *i* is the number of sample mass measurements of the second stage, *N* is the number of measurements of the second stage, and $M^{F}\left(t_{i}\right)$ is the approximating function according to the Fick diffusion law of the first stage. As a result, the maximum moisture content of plastics is estimated as the sum of the approximated value of the equilibrium moisture content in the first stage and an estimate of the growth jump in the second stage (Equation 4) in the following form: (4)$$M_{\infty }=M_{1}+\Delta M.$$

Results of approximation of the sorption–desorption kinetics of RP are presented in Table 4. The increases in the average moisture content in the second stage ΔM after 2 years and 4 years of exposure to a cold climate are 258% and 517% for GRP and 14% and 23% for BRP; meanwhile, the increase in the average moisture content in the second stage ΔM of the unexposed GRP sample is 4.6 times greater than that of the unexposed BRP sample. The maximum moisture content for GRP increases from 0.35% to 0.54% after exposure to 4 years of cold climate due to an increase in moisture content in the second stage. For GRP, this increases from 0.38% to 0.46% after 4 years of exposure to the cold climate due to the increase in the moisture content of the first stage.

For BRP, there is a post-curing process; for GRP, in addition to the post-curing, there is also a process of destruction during exposure; additionally, note that the BRP sample has a large number of technological pores and that the exposed GRP sample has a large amount of delamination of the polymer matrix from the reinforcing fabric. Taking these notes into account, it can be concluded that the first stage of moisture absorption kinetics is associated with the transverse sorption of moisture into the sample; the second stage of kinetics is associated with the longitudinal sorption of moisture into the sample, causing further detachment of the polymer matrix from the reinforcing fabric under the influence of thermal and humidity factors. Thus, the anomalous diffusion of moisture can be observed [26]. An anomalous moisture diffusion of the same type was observed in the work of [27].

### 3.5. Monitoring IR Absorption Regions of RPs

Figure 9 and Figure 10 show the IR spectra for BRP and GRP samples left unexposed and exposed for 2 and 4 years.

In samples subjected to climatic tests, according to IR spectrometry data, a wide absorption band appears in the region of 3100–3550 cm^−1^; this is most likely associated with the effect of atmospheric moisture and belongs to hydroxyl OH groups linked by hydrogen bonds. Hydrogen bonds contribute to intense absorption in the region of 2365 cm^−1^ and 2336 cm^−1^, which relate to the absorption of dimers according to [28].

Visual representations of the partial destruction of the samples after the climatic tests—according to their IR spectra—are displayed, represented by the increase in the intensity of the absorption bands of the methylene groups at 1368, 1352, and 1304 cm^−1^ and by the intensity of the high-frequency band in the region of 1300–1400 cm^−1^; this indicates an increase in the content of terminal methyl groups.

The peak at 1650–1660 cm^−1^ becomes more intense with the predominance of the resonance effect of the substituent with a wide band in the region of protogenic groups 3100–3550 cm^−1^; this may indicate the presence of a carbonyl group, leading to a change in the color of the polymer matrix [29,30,31], hydrogen bonded to two or more adsorbed water molecules.

The post-curing of the BP and SP sample in the first years can be observed in the IR spectrum by an increase in the spectral line of 1.733 cm^−1^; this is because, according to [32], the spectral line of 1.733 cm^−1^ fully characterizes the cross-linking of the network structure of the epoxy polymer using ester bonds. This concerns the ruptures of the mesh structure in the form of carboxyl groups; then, the stretching vibration of the C = O group in its composition (spectral line 1710 cm^−1^) serves as an indicator of their presence.

No significant changes in the spectral peaks (appearance of new ones or disappearance of existing ones) corresponding to other functional groups of the polymer matrix are observed; this leads to the conclusion that the polymer matrix is relatively inert, that it does not enter into chemical reactions, or that its rate is very low.

## 4. Conclusions

For a reliable assessment of the climatic resistance of PCMs, the laws of the initial stage of their climatic aging should be taken into account. This includes the post-curing of the polymer matrix and the relaxation of the technological nonequilibrium state, which lead to an increase in the elastic strength properties of the materials. Therefore, as a starting point in investigating the climatic aging of PCMs, it is necessary to accept the time of exposure; here, the maximum values of the elastic strength properties of polymeric materials are achieved. With this approach, relative to 2 years of exposure, after 4 years of exposure, the tensile strength for stretching decreases by 15% in BRP and in GRP it decreases by 22%; the limit of strength during the bend decreases by 12% in BRP and by 47% in GRP.

Based on the results of the DMA analysis, it was found that, unlike BRP (where the material is post-cured exclusively at the initial stage of the exposure), a process of destruction occurs in the GRP. The formation of internal stresses in the material and its growth from the duration of climatic exposure was established.

The presence of two-stage kinetics of moisture sorption was revealed: growth according to the Fickian diffusion law and a jump in moisture content. The anomaly of moisture diffusion in BRP and GRP is associated with the distribution of closed porosity during manufacturing, as well as the hygroscopicity of the fibers. Despite the fact that the initial porosity of GRP is less than that of BRP, the stronger hygroscopicity of fiber glass compared to fiber basalt (for glass fiber it is 10–20%; it is less than 1% for basalt fiber) led to the fact that the maximum moisture content of GRP turned out to be greater than that of BRP, especially after 4 years of exposure to cold climates. The formation of closed porosity, depending on the duration of exposure, can be assessed using the values of the increase in the average moisture content in the second stage: BRP (0.09%); SRP (0.3%). The obtained result is correlated with the results of IR spectrometry and fractography.

A set of experimental studies has established that GRP is subject to greater destruction under the influence of a very cold climate than BRP. The results obtained can be used to improve the scientific validity of the use of basalt plastics and reduce costs in preparation for and performance of climatic tests.

## Figures and Tables

**Figure 1 polymers-16-00866-f001:**
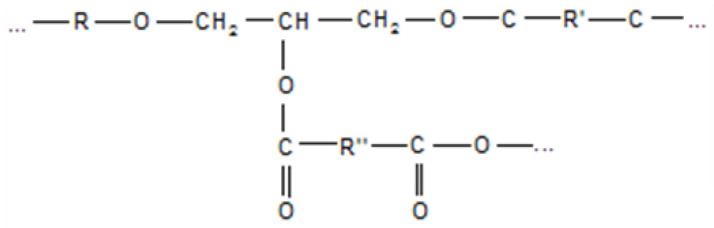
The spatial network of a polymer matrix during anhydride curing.

**Figure 2 polymers-16-00866-f002:**
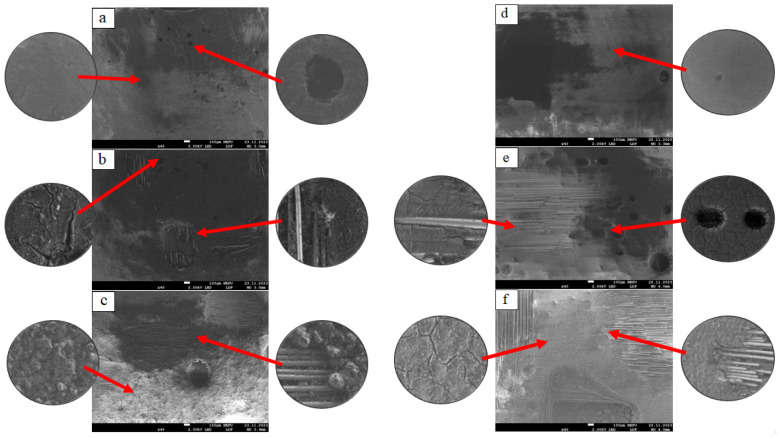
Microstructureof the solar surface of reinforced plastis: (**a**) unexposed BRP; (**b**) exposed for 2 years BRP; (**c**) exposed for 4 years BRP, (**d**) unexposed GRP; (**e**) exposed for 2 years GRP; (**f**) exposed for 4 years GRP.

**Figure 3 polymers-16-00866-f003:**
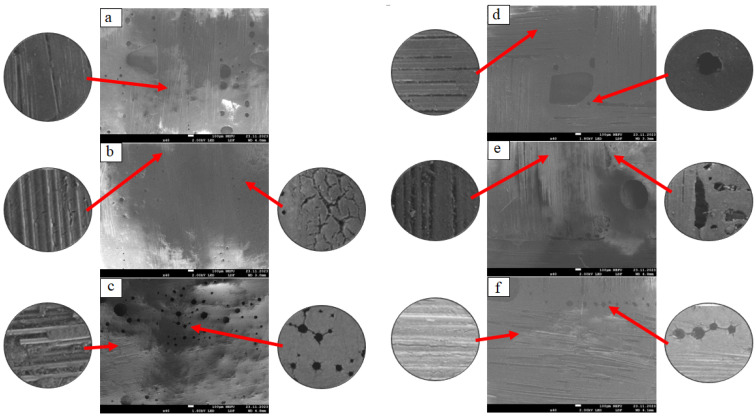
Microstructure of the shadow surface of reinforced plastis: (**a**) unexposed BRP; (**b**) exposed for 2 years BRP; (**c**) exposed for 4 years BRP, (**d**) unexposed GRP; (**e**) exposed for 2 years GRP; (**f**) exposed for 4 years GRP.

**Figure 4 polymers-16-00866-f004:**
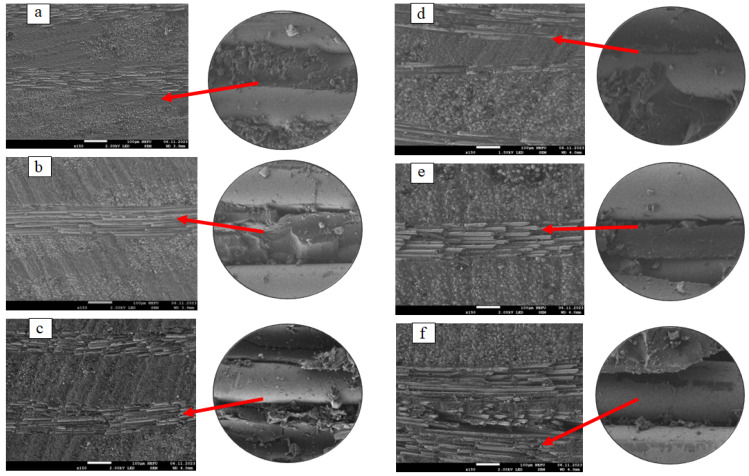
Microstructure of BRP and GRP sections: (**a**) unexposed BRP; (**b**) exposed for 2 years BRP; (**c**) exposed for 4 years BRP, (**d**) unexposed GRP; (**e**) exposed for 2 years GRP; (**f**) exposed for 4 years GRP.

**Figure 5 polymers-16-00866-f005:**
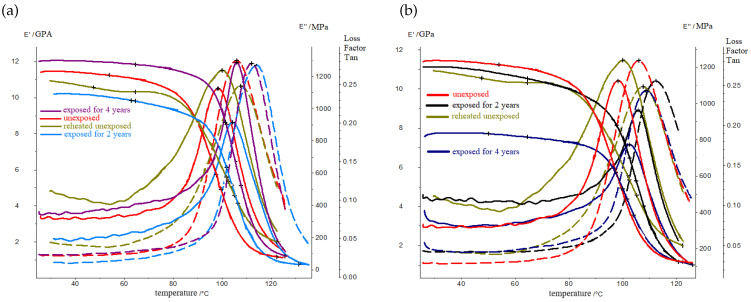
Elastic modulus, loss modulus, and mechanical loss tangent of the solar- and shadow-exposed parts of the BRP samples: (**a**) the solar-exposed side; (**b**) the shadow-exposed side.

**Figure 6 polymers-16-00866-f006:**
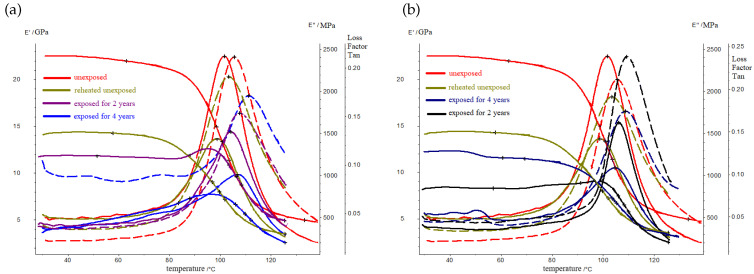
Elastic modulus, loss modulus, and mechanical loss tangent of the solar- and shadow-exposed parts of the GRP samples: (**a**) the solar-exposed side; (**b**) the shadow-exposed side.

**Figure 7 polymers-16-00866-f007:**
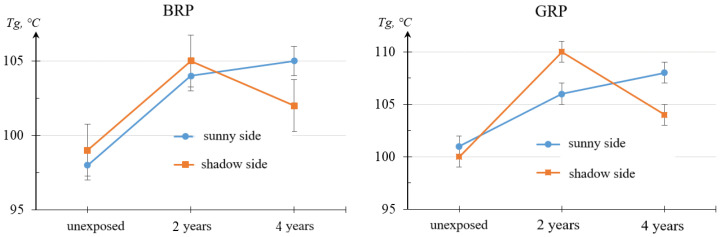
Glass-transition temperature of RPS samples which were unexposed, exposed for 2 years, and exposed for 4 years: BFR and GRP.

**Figure 8 polymers-16-00866-f008:**
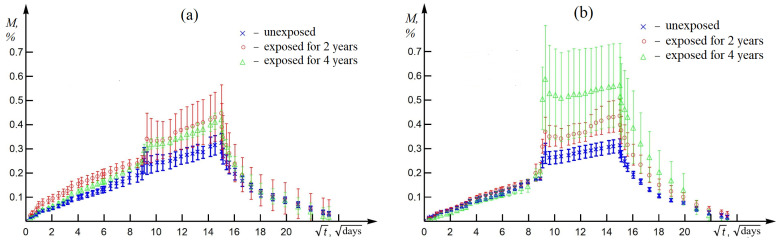
Kinetics of sorption at 23 °C/RH 68% and moisture desorption at 23 °C in RP samples: (**a**) BRP; (**b**) GRP.

**Figure 9 polymers-16-00866-f009:**
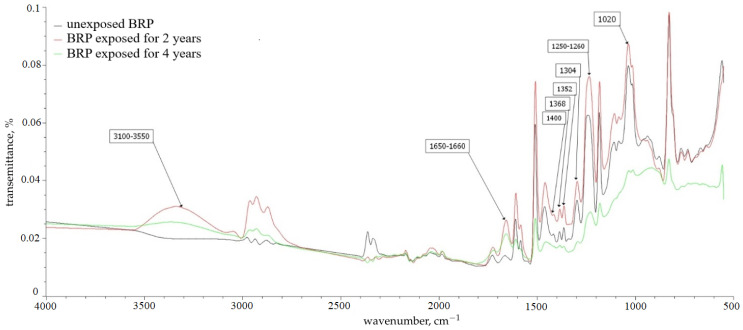
IR spectra of BRP samples.

**Figure 10 polymers-16-00866-f010:**
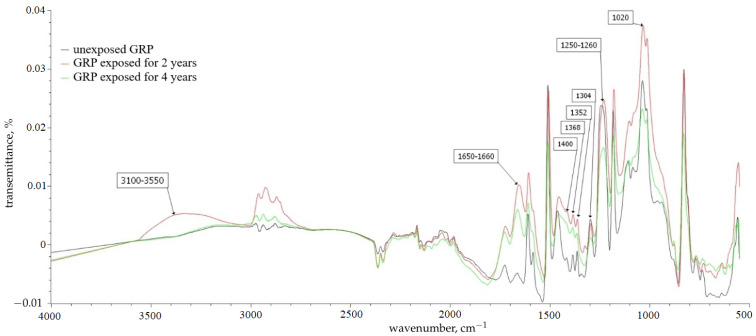
IR spectra of GRP samples.

**Table 1 polymers-16-00866-t001:** A relative retention coefficient, $k_{R}$, of RPs.

RP	*R*	$k_{R}$, t=2 Years	$k_{R}$, t=4 Years
BRP	tensile strength	1.04	0.89
BRP	bending strength	1.43	1.31
GRP	tensile strength	1.11	0.89
GRP	bending strength	1.56	1.09

**Table 2 polymers-16-00866-t002:** Characteristic temperatures of the DMA analysis of the solar- and shadow-exposed sides of the BRP.

Indicator Name	Unexposed	Unexposed	Solar-Exposed Side (2 Years)	Shadow-Exposed Side (2 Years)	Solar-Exposed Side (4 Years)	Shadow-Exposed Side (4 Years)
The beginning of the transition on the elastic modulus curve—$T_{onset}$, (°C)	86	86	95	96	96	93
Inflection point—$T_{g}$, (°C)	98	99	104	105	105	102
The end of transition on the elastic modulus curve—(°C)	107	111	114	112	114	110
Peak on the tangent curve—$T_{tg}$, (°C)	106	107	114	112	112	109
Peak in loss modulus curve—$T_{loss}$, (°C)	98	100	104	106	106	102
Thickness—mm	3.67	2.10	2.43	3.50	3.85	2.55
Sample weight before testing—g	2.912	1.589	1.871	2.691	2.901	1.944
Sample weight after testing—g	2.908	1.587	1.869	2.689	2.898	1.942
Weight loss percentage—%	0.138	0.126	0.107	0.074	0.104	0.103

**Table 3 polymers-16-00866-t003:** Characteristic temperatures of the DMA analysis of the solar- and shadow-exposed sides of the GRP.

Indicator Name	Unexposed	Unexposed	Solar-Exposed Side (2 Years)	Shadow-Exposed Side (2 Years)	Solar-Exposed Side (4 Years)	Shadow-Exposed Side (4 Years)
The beginning of the transition on the elastic modulus curve—$T_{onset}$, (°C)	90	84	98	99	102	92
Inflection point—$T_{g}$, (°C)	101	100	106	110	108	104
The end of transition on the elastic modulus curve—(°C)	110	111	116	117	115	114
Peak on the tangent curve—$T_{tg}$, (°C)	106	103	108	111	109	109
Peak in loss modulus curve—$T_{loss}$, (°C)	102	99	104	107	106	105
Thickness—mm	3.30	2.21	2.27	3.48	3.05	2.87
Sample weight before testing—g	3.021	1.776	1.900	2.996	2.500	2.203
Sample weight after testing—g	3.020	1.775	1.898	2.991	2.493	2.200
Weight loss percentage—%	0.033	0.056	0.105	0.167	0.281	0.136

**Table 4 polymers-16-00866-t004:** Results of approximation of the sorption–desorption kinetics of RP.

RP	*D*	$M_{1}$	ΔM	$M_{\infty }$
Unexposed BRP	20.5	0.39	0.003	0.39
Exposed BRP for 2 years	29	0.49	0.05	0.49
Exposed BRP for 4 years	27	0.36	0.09	0.45
Unexposed GRP	23.7	0.32	0.047	0.37
Exposed GRP for 2 years	23.1	0.33	0.14	0.47
Exposed GRP for 4 years	25.2	0.24	0.3	0.59

## Data Availability

The data used to support the findings of this study are available from the corresponding author upon request.

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
