# Peer review of "The Initial Stage of Climatic Aging of Basalt-Reinforced and Glass-Reinforced Plastics in Extremely Cold Climates: Regularities"

_polymers, 2024, doi:10.3390/polym16070866_

Round 1

Reviewer 1 Report

Comments and Suggestions for Authors

In this paper, climatic aging effect of the extremely cold climate on physical properties of basalt-reinforced plastic and glass-reinforced plastic in were investigated. It is of great significance for the application of these two kinds of materials.

Comments:

1. In section 3.1, the exposure times are 12 months and 24 months. Those are diferent from the exposure times of 2 years and 4 years in other sections. Why not to unify the exposure times? And the time points are very little. I think four or five is OK for this paper.

2. The curves in Fig. 5 and Fig. 6 are messy. The title of those two figures does not match the content in the figures. There are not (a), (b) and (c) in those two figures. And also Russian appear in those two figures.

3. The reason for the great difference between the curves of the two unexposured specimens In Fig. 5 and Fig. 6 should be given in detail.

4. The mass change of Fig. 8 is different from typical water sorption curves. Please check the test results carefully again.

Author Response

Responses to your comments are provided in the attached file.

Reviewer 2 Report

Comments and Suggestions for Authors

Why does GFRP degrading faster than BRP in cold climates?

The authors of this manuscript stated their discovery eloquently. But they did not provide a convincing reason for GFRP degradation rate being higher than BRP degradation arte especially under bend (as they call). The authors need to explain in full the degradation phenomenon of GFRP over BRP before publishing this manuscript.

Author Response

(The authors gave the same response as above.)

Round 2

Reviewer 1 Report

Comments and Suggestions for Authors

The paper has been revised and is suggested to be accepted.